# Preparation and Properties of Cassava Residue Cellulose Nanofibril/Cassava Starch Composite Films

**DOI:** 10.3390/nano10040755

**Published:** 2020-04-15

**Authors:** Lijie Huang, Hanyu Zhao, Tan Yi, Minghui Qi, Hao Xu, Qi Mo, Chongxing Huang, Shuangfei Wang, Yang Liu

**Affiliations:** 1College of Light Industry and Food Engineering, Guangxi University, Nanning 530004, China; hannaxi08@163.com (H.Z.); YITANgxu@163.com (T.Y.); mhqi9812@163.com (M.Q.); 17088743942@163.com (H.X.); MoQiGX@163.com (Q.M.); 2Guangxi Key Laboratory of Clean Pulp & Papermaking and Pollution Control, Nanning 530004, China; huangcx21@163.com (C.H.); wshuangfei@163.com (S.W.); xiaobai@gxu.edu.cn (Y.L.)

**Keywords:** cassava residue, cellulose nanofibril, modified, cassava starch, composite film, nanocomposite

## Abstract

Because of its non-toxic, pollution-free, and low-cost advantages, environmentally-friendly packaging is receiving widespread attention. However, using simple technology to prepare environmentally-friendly packaging with excellent comprehensive performance is a difficult problem faced by the world. This paper reports a very simple and environmentally-friendly method. The hydroxyl groups of cellulose nanofibrils (CNFs) were modified by introducing malic acid and the silane coupling agent KH-550, and the modified CNF were added to cassava starch as a reinforcing agent to prepare film with excellent mechanical, hydrophobic, and barrier properties. In addition, due to the addition of malic acid and a silane coupling agent, the dispersibility and thermal stability of the modified CNFs became significantly better. By adjusting the order of adding the modifiers, the hydrophobicity of the CNFs and thermal stability were increased by 53.5% and 36.9% ± 2.7%, respectively. At the same time, the addition of modified CNFs increased the tensile strength, hydrophobicity, and water vapor transmission coefficient of the starch-based composite films by 1034%, 129.4%, and 35.95%, respectively. This material can be widely used in the packaging of food, cosmetics, pharmaceuticals, and medical consumables.

## 1. Introduction

The emergence of plastic products has greatly facilitated human life [1]. To date, plastic products have appeared in almost every corner of the globe. Data show that the annual output of plastics increased from 1.5 million tons in the 1950s to 359 million tons in 2018 [2]. The huge amount of plastic produced, because of its inability to degrade, has caused “white pollution” that seriously threatens the Earth’s ecological environment [3]. With the temperature rise at the north and south poles and epidemics raging, protecting the environment and ecosystems has become an urgent need. In the search for alternative materials for plastic products, starch and cellulose have become the best alternatives due to their non-toxicity, degradability, low cost, and wide availability [4,5,6].

Starch film is transparent, edible, safe, resistant against folding, and is widely used in the preparation of biodegradable blends and composite materials. Starch is considered one of the most promising green packaging materials [7,8]. However, pure starch film has the disadvantages of a high brittleness, low mechanical properties, and poor water blocking performance, which limit its further development [9,10]. In recent years, to improve the strength and barrier properties of starch films, researchers have often added different types of enhancers to starch film to improve its strength [11,12,13,14]. Cellulose nanofibril (CNF) has become an ideal starch film enhancer due to its low cost, low density, renewability, recyclability, high surface area, chemical reactivity, strength, modulus, elasticity, transparency, tensile rigidity, light weight, low thermal expansion, and biodegradability (due to its nano-size characteristics) [15,16,17].

Cellulose nanofibril comes from various sources of natural fibers, such as cotton, wood, corn cobs, sisal, wheat straw, flax, bamboo, rice husks, pea husks, coconut shells, bagasse, and cassava residues. However, CNF is hydrophilic and absorbs moisture when exposed [18]. Therefore, the surface hydrophobicity of CNF can be changed using various chemical modification techniques, thereby improving the compatibility and dispersibility of CNF in specific solvents [19]. Through phosphorylation, carboxymethylation, oxidation and sulfonation reactions, ionic charge can be introduced to the surface of cellulose [20,21,22,23]; esterification, silylation, amidation, urethanation, and etherification can make the cellulose surface hydrophobic [24,25,26]. In summary, no matter what surface chemistry is ultimately required, the modification technology depends almost entirely on the reaction of the hydroxyl groups on the surface of the CNF. The challenge for these chemical modification technologies is to change only the surface of the CNF, maintaining its original morphology and the complex structure of its internal hydroxyl groups.

Wei et al. extracted CNF from oil palm waste by acid hydrolysis using natural lime juice as a cross-linking agent. The structure, stretch performance, transparency, color, and water vapor transmission rate of these composite materials prepared by CNF-filled cassava starch were then studied. When lime juice was used as a cross-linking agent, the tensile strength of the composite film increased with the CNF content. Moreover, the composite film was found to be more elastic and have higher elongation at break, significantly improved transparency, and reduced moisture absorption [27]. Fitch-Vargas et al. explored the influence of acetylated fiber and glycerin content on the mechanical, physical, and microstructural properties of acetylated corn-starch-based biocomposites. The acetylated sugarcane-fiber-filled starch-based composite prepared by thermoplastic extrusion was found to have favorable mechanical properties and water resistance [28]. Ibrahim et al. prepared biodegradable composite membranes using thermoplastic corn starch and corn husk fiber as reinforcing fillers. The tensile strength, Young’s modulus, crystallinity, thermal stability, and initial decomposition temperature of the composite were improved because of the fiber addition, although the tensile strength did not increase above 12.84 MPa [29]. The use of microfibrillated cellulose, bamboo fiber, bagasse fiber, cotton fluff fiber, and sisal fiber [30,31,32,33] in reinforcement-filled starch-based composite films has been widely studied. However, research on using cassava residue CNF as reinforcement to fill cassava starch-based films is very limited.

Cassava residue has a high fiber content, good thermal stability, and high crystallinity, which has aroused research interest in this subject. This study modified the CNFs of cassava residue. l-malic acid and the silane coupling agent KH-550 were used for esterification (Z-CNF), cross-linking (J-CNF), cross-linking followed by esterification (J-Z-CNF), and esterification followed by cross-linking (Z-J-CNF). The effects of different modification methods and modification sequences on the structure and properties of the resulting cassava residue CNFs were then studied. Different quantities of CNFs and modified CNFs were added to thermoplastic cassava starch (TPS) film and their effects on the mechanical properties, hydrophilicity, water vapor barrier, oil permeability, water solubility, and roughness of the composite films were studied. The prepared composite films have excellent mechanical properties, hydrophobicity, and barrier properties. Not only does it greatly alleviate the environmental pollution caused by a large amount of cassava residue being discarded, but it also provides a certain theoretical basis for the further development and utilization of cassava residue.

## 2. Experimental Section

### 2.1. Materials

Cassava residue (Guangxi Wuming County Anning Starch Co., Ltd., Nanning, China), γ-aminopropyl triethoxysilane, the silane coupling agent KH-550 (97%, Shandong Yousuo Chemical Technology Co., Ltd., Linyi, China), l-malic acid (98%, Shanghai Aladdin Biochemical Technology Co., Ltd., Shanghai, China), sodium bisulfate monohydrate (≥98.5%, Tianjin Bodi Chemical Co., Ltd., Tianjin, China), potassium bromide (analytical grade, Chengdu Jinshan Chemical Reagent Co., Ltd., Chengdu, China; phosphotungstic acid (Tianjin Damao Chemical Reagent Factory, Tianjin, China), and medium temperature α-amylase (food grade, Beijing suolaibao Technology Co., Ltd., Beijing, China) were used in this study.

### 2.2. Preparation of CNF

First, the cassava residue was washed, dried, and sifted, and then dried to a constant weight. Then, 50 g of cassava residue was weighed into a beaker, 500 mL of distilled water was added, and the mixture was heated to a constant temperature of 95 °C in a water bath for 30 min to completely gelatinize the starch in the cassava residue. Subsequently, the temperature was decreased to 60 °C, the pH of the solution was adjusted to 5.8 with 1% sodium hydroxide, and 25 mL of a previously prepared 1% α-amylase solution was added into the beaker, allowing enzymatic hydrolysis to occur for 3 h, while the temperature and pH were maintained during this process. After that, the enzyme was deactivated by heating at 100 °C for 5 min, and the cassava residue was washed with distilled water until a neutral pH was reached. Then, the cassava residue was filtered and dried at 65 °C for 8 h. Subsequently, the fibers were treated with 1% (w/v) sodium chlorite to bleach them, and the solution pH was maintained at 4–5 with the addition of acetic acid, aiming for a cellulose-to-liquid ratio of 1:20 (g/mL). The bleaching treatment was performed again at 70 °C for 2 h using the same amount of sodium chlorite and acetic acid as before. After that, the bleached cellulose was washed with distilled water until a neutral pH was reached, and then dried at 50 °C for 12 h. After amylase pretreatment and bleaching, CNFs were prepared by homogenizing the fibers using a high-pressure micro jet homogenizer. The homogenization pressure was set to 15,000 psi, and the sample was homogenized 30 times. The sample suspension with an expected CNF measured mass fraction of ~0.392% was collected for analysis.

### 2.3. Modification of CNF

To prepare J–CNF, 0.16 mL of the silane coupling agent, KH-550 was added to 100 mL of distilled water and stirred on a magnetic stirrer for 30 min to fully hydrolyze KH-550, and then 77 mL of the CNF suspension with a mass fraction of 0.392% were added. After mixing, the mixture was sonicated at 1000 W for 45 min, and then centrifuged and washed with distilled water to remove any unreacted silane coupling agent. The sample was centrifuged five times at a speed of 10,000 r/min for 10 min each to obtain J-CNF.

To prepare Z-CNF, 256 mL of the CNF suspension with a mass fraction of 0.392% (the dry weight of CNF was 1.00 g) were placed in a three-necked flask and heated in an oil bath to 110 °C, and then 10 g of L-malic acid along with 0.1 g of sodium bisulfate monohydrate were added, and the reaction mixture was heated continuously. After 8 h, the unreacted L-malic acid was removed through centrifugation and washing with distilled water. The reaction mixture was centrifuged five times at 10,000 r/min for 10 min to obtain Z-CNF.

Further, J-CNF was esterified with L-malic acid following the same procedure described above to obtain J-Z-CNF.

Finally, the silane coupling agent KH-550 was used to cross-link Z-CNF following the procedure described for obtaining J-CNF. The so-obtained product is denoted as Z-J-CNF.

### 2.4. Preparation of Cassava Residue CNF/Cassava Starch Composite Film

Cassava starch (5 g), a known quantity of modified CNF (0, 0.1, 0.2, 0.3, 0.4, or 0.5 g), and 1.5 g of glycerol were added to 100 mL distilled water. The resulting mixture was stirred well and heated in a water bath at 90 °C for 30 min under stirring. Then, the solution was degassed by sonicating at room temperature for 20 min. Subsequently, 40 mL of the obtained film-forming solution were poured into a square plastic Petri dish (side length, 12 cm), and dried at 50 °C for 8 h. Films of pure starch and unmodified CNF/TPS composite without any additional enhancer were also prepared in the same manner for comparison. Before testing, the films were peeled from the molds and stored at 50% relative humidity and 23 °C ± 1 °C for 2 days. The film thickness was determined to be approximately 0.1 mm.

### 2.5. Testing and Characterization

#### 2.5.1. Apparent Morphological Representation

To characterize the morphology and size of CNF samples, transmission electron microscopy (TEM; Hitachi HT7700, Hitachi High-tech Co., Ltd., Shenzhen, China) was performed at an acceleration voltage of 80 kV. Prior to observation, the CNF suspension was diluted to a concentration of 0.008% with distilled water and mixed magnetically for 4 h to ensure an even dispersion of the CNFs in water. Then, a drop of the suspension was cast on a copper grid. After the water evaporated, a drop of a solution of phosphotungstic acid dye was placed on the copper grid to dye the CNFs. After dyeing for 0.5 h, filter paper was used to remove excess dye.

Atomic force microscopy (AFM; Hitachi 5100N, Hitachi High-tech Co., Ltd., Shenzhen, China) was conducted in the tapping mode for additional sample characterization. The test pressure was 0.55 Hz and the elastic constant of the cantilever was 26 N/m.

#### 2.5.2. Characterization of the Chemical Structure

Fourier-transform infrared spectrometry (FT-IR; TENSOR II, BRUKER, Billerica, MA, USA) was performed over the wavenumber range of 400–4000 cm^−1^ with a resolution of 4 cm^−1^ to characterize the CNFs before and after modification.

X-ray photoelectron spectroscopy (XPS; ESCALAB 250XI+, Thermo Fisher Scientific, Waltham, MA, USA) was conducted to characterize the changes in the surface elements and chemical functional groups of CNFs before and after modification with the silane coupling agent KH-550. The spot diameter was 500 µm.

#### 2.5.3. Characterization of the Crystallinity

The crystallinity of the CNFs was determined by X-ray diffraction (XRD) at room temperature. The CNF and modified CNF samples used for XRD were crushed and sieved through a 100-mesh sieve, and then dried in an oven at 65 °C. The scanning range was 2*θ* = 5°–50° and the scanning speed was 5°/min. The crystallinity of the sample was calculated as follows:(1)$$CrI\;(\%)=\frac{I_{002}-I_{am}}{I_{002}}\times 100$$
where *CrI* is the crystallinity index; *I*_002_ is the maximum diffraction peak intensity, representing the crystallization region and amorphous region; and *I*_am_ (2*θ* = 18°) is the diffraction peak intensity, representing the amorphous region.

#### 2.5.4. Thermal Stability Analysis

A synchronous thermal analyzer (STA4 49F5, NETZSCH, Selb, Germany) was used to analyze the thermal weight loss of the samples to determine their thermal stability. Before analysis, the sample was crushed and sieved through a 100-mesh sieve and dried to a constant weight. The test was conducted under a N_2_ gas environment at a gas flow rate of 20 mL/min in the temperature range of 25–700 °C at a heating rate of 10 °C/min.

#### 2.5.5. Characterization of Hydrophobic Properties

The hydrophilicity of pure CNF and modified CNFs was studied using a Clus contact angle measuring instrument (DSA100, Phenom Company, Delft, The Netherlands). The composite films prepared from pure CNF and modified CNFs were cut into slender specimens (10 × 5 mm) and adhered to glass supports using double-sided adhesive tape. Each slide was placed on the stage of the contact angle measuring instrument and the contact angle of a 3-μL water droplet on the composite film was measured.

For water absorption studies, the film samples were cut into 25 mm × 20 mm specimens, dried in an oven at 80° C to a constant weight, and then placed in a desiccator with discolored silica gel and saturated calcium nitrate solution for one week. Three samples of each film were collected, and their average water absorption rates were determined using Equation:(2)$$\mathrm{Water}\;\mathrm{absorption}\;\mathrm{rate}=\frac{W_{t}-W_{0}}{W_{0}}\times 100\%$$
where *W_t_* is the weight of the wet film sample at time *t* and *W*_0_ is the weight of the dry film sample.

#### 2.5.6. Characterization of Mechanical Properties

An electronic universal material testing machine (3367, American Instron Corporation, Norwood, MA, USA) was used to evaluate the mechanical properties of the nanocomposite films. The initial distance and tensile speed of the fixture were 50 mm and 1 mm/s, respectively. The tests were repeated at least five times for each sample, and the average values of these test results are reported.

#### 2.5.7. Barrier Characterization

A water vapor transmission rate tester (PERMATRAN-W Model 3/61, American MOCON Company, Minneapolis, MN, USA) was used to determine the moisture permeability of the composite film according to the GB/T 26253-2010 standard.

A tube containing 5 mL of soybean oil was sealed with a small film of 40 mm × 40 mm dimensions, and the tube was inverted on a medium speed qualitative filter paper in a dryer for 3 days, after which the weight change of the filter paper was recorded. The reported oil permeability (OP; gmm·m^−2^·d^−1^) coefficient is the average of three samples per film, which was calculated as follows:(3)$$OP=\frac{\Delta W\times d}{s\times t}$$
where Δ*W* is the change in the weight of the filter paper; *S* is the area of the film (m^2^); *d* is the thickness of the film (mm); and *t* is the number of days of storage.

## 3. Results and Discussion

### 3.1. Testing and Characterization of Modified CNF

#### 3.1.1. Apparent Morphological Representation

Figure 1a–e shows TEM images of CNF, J-CNF, Z-CNF, J-Z-CNF, and Z-J-CNF, respectively. Compared with the transmission electron micrograph of the CNF, the dispersion of the modified CNFs is evidently better. The three-dimensional network structure of the original CNF is maintained between the filaments, with good separation. There is no large CNF aggregate in the TEM diagram, only the crosslinked structure formed in the drying process. The main reason for this is that the esterification of l-malic acid with the surface hydroxyl group of the CNF reduces the hydroxyl number on the surface of the CNF, avoiding the formation of hydrogen bonds to some extent, and reducing the probability of agglomeration between the CNFs [34]. In addition, because the silane coupling agent hydrolyzes to form silanol in water, the –OH on the surface of silanol and the –OH on the surface of the CNF undergo hydrogen bonding. As a result, KH-550 covers the CNF well, and the dispersibility and stability of the CNF were improved [35].

Comparing Figure 1b–e, the dispersion of Z-CNF and J-Z-CNF is significantly better than that of J-CNF and Z-J-CNF. The main reason for this result is that KH-550 relies on replacing hydrogen bonds and encapsulation to improve the dispersibility in a weaker way than l-malic acid improves the dispersibility through esterification. Moreover, it is difficult for a small amount of KH-550 to effectively enter the interior of CNF aggregates and effectively wrap the CNFs. l-malic acid can be directly grafted onto the surface of the CNFs and directly reduces the hydrogen bonding between the CNFs, which is also the main reason for the large number of meshes in Figure 1c, d. It is worth considering that, although l-malic acid plays a role, the dispersion of the CNF in Figure 1e is not high; a possible reason for this is that the addition of KH-550 in the second step made the dispersed CNFs re-aggregate under the action of encapsulation and coupling [36,37].

Figure 2a–e shows the AFM images of CNF, J-CNF, Z-CNF, J-Z-CNF, and Z-J-CNF, respectively; the phase and the topographic images of the modified CNFs are arranged from left to right in each image. Compared with the AFM data for pure CNF, Figure 2b shows that the diameter of the cross-linked CNF, J-CNF, is larger, while Figure 2c shows that the diameter of the CNF esterified with malic acid becomes smaller, and Figure 2d, e shows no obvious changes in the diameter of J-Z-CNF and Z-J-CNF. KH-550 generates silanol and silanol oligomer through hydrolysis and polymerization reactions. The hydroxyl groups on the surface of both the silanol and the CNF are hydrogen or chemically bonded, which is akin to introducing grafted molecular chains on the CNF surface, and the diameter of J-CNF becomes larger as a result. In addition, the wrapping effect of KH-550 on the CNF may prompt multiple CNFs to be wrapped at the same time, resulting in an increase in diameter [35,38]. The smaller diameter of Z-CNF can be attributed to the surface peeling effect of highly grafted cellulose or degradation of natural cellulose under acidic reaction conditions [39].

#### 3.1.2. Chemical Structure

Figure 3 shows that J-CNF has new characteristic FT-IR peaks at 1582 and 789 cm^−1^, which can be attributed to KH-550. The absorption peak at 1582 cm^−1^ is due to the bending of –NH_2_ and that at 789 cm^−1^ is due to the out-of-plane bending of N–H. In theory, in the range of 1000–1200 cm^−1^, there should be absorption peaks of Si–O–Si bonds formed by the hydrolysis of KH-550 and Si–O–CH_3_ bonds formed by the reaction of KH-550 and CNF, but it is difficult to identify the characteristic absorption peaks of Si–O–Si and Si–O–CH_3_ because of the strong absorption peaks of CNF in the same region [35,40]. At the same time, the CNF peak associated with hydrogen bonding is weakened after modification. This is mainly due to the weakening of hydrogen bonds between and within the CNF after modification. In addition, the –OH absorption peak in the spectrum curve of J-CNF is enhanced. This is the result of the coupling agent acting instead of the original hydrogen bond.

The FTIR spectrum of Z-CNF in Figure 3 shows new characteristic absorption peaks at 2967, 1735, and 1258 cm^−1^. The peak at 2967 cm^−1^ is the asymmetric stretching vibration peak of CH_3_, the peak at 1735 cm^−1^ is the C=O stretching vibration peak of the ester group, and the peak at 1258 cm^−1^ is the C–O stretching vibration peak of the ester group. These signals indicate that malic acid was successfully grafted onto CNF [39]. The infrared spectra of J-Z-CNF and Z-J-CNF are very similar, and there are characteristic absorption peaks related to the coupling agent and malic acid at 2967 cm^−1^ (CH_3_), 1735 cm^−1^ (C=O), 1258 cm^−1^ (C–O), 1582 cm^−1^ (–NH_2_), and 789 cm^−1^ (N–H), which indicate that the sample was successfully esterified.

In these infrared spectra, the characteristic peaks of the silane coupling agent-modified CNF are not obvious. Therefore, the samples were further characterized by XPS. As seen in the wide-scan spectra of CNF and J-CNF in Figure 4a, both C1s and O1s peaks appeared for CNF and J-CNF. Three new electron peaks appeared in the wide-scan spectrum of J-CNF: a peak of N1s at 398 eV, a peak of Si2p at 101.17 eV, and a peak of Si2s at 152.3 eV. Figure 4b shows a high-resolution XPS spectrum of the Si species of J-CNF, which confirms the modification of CNF by the silane coupling agent KH-550. The peaks of Si–O–Si (101.4 eV) and SiO_3_ (102.4 eV) bonds indicate that the coupling agent [NH_2_–(CH_2_)_3_–Si(OC_2_H_5_)_3_] was hydrolyzed to form silanol [NH_2_–(CH_2_)_3_–Si(OH)_3_]. The silanol or partially condensed silanol was coated onto the CNF surface [40].

The carbon atoms of CNF exist in three chemical states, viz. C1, C2, and C3, where C1 is in the form of C–C or C–H, C2 is in the form of C–OH, and C3 is in the form of O–C–O [41]. As shown in Table 1, compared with the proportions of carbon states in CNF, the proportion of C1 in J-CNF was increased to 33% ± 2%, which is caused by the propane group of the silane coupling agent and the proportion of C2 was decreased to 38% ± 3%, which indicates that the hydrophilicity of CNF decreases because the hydroxyl reaction between KH-550 and CNF reduces the number of hydroxyl groups on the surface of the CNF.

#### 3.1.3. Crystallinity

Similar to that of the original CNF, the XRD patterns of J-CNF, Z-CNF, J-Z-CNF, and Z-J-CNF (Figure 5) show three main diffraction peaks at approximately 2*θ* = 16.2°, 22°, and 34.6°, without any change in the position of the diffraction peaks. Based on this, silanization and esterification did not change the crystal structure of CNF, which remained cellulose type I after processing.

Table 2 shows the crystallinity before and after the modification of CNF. The results show that the crystallinity of the modified CNFs is lower than that of unmodified CNF, and the crystallinities of J-CNF, Z-CNF, J-Z-CNF, and Z-J-CNF are 53.2% ± 4.2%, 55.4% ± 4.3%, 53.1% ± 2.8%, and 50.3% ± 3.1%, respectively. Compared with that of CNF, the crystallinity of J-CNF and Z-CNF does not decrease significantly. This is because the modification of CNF by the silane coupling agent KH-550 is a surface modification process. Strong hydrogen bonds were formed between the silanol groups and hydroxyl groups on the surface of CNF, destroying the hydrogen bonds between the CNF molecules, which resulted in a decrease in the crystallinity of CNF [40]. Further, the modification of CNF by l-malic acid only takes place on the surface of CNF and does not damage the internal structure of CNF [42]. In addition, compared with the crystallinity, the CNF crystallinity after the coupling agent modification and the double modification is greatly reduced. The main reason for this reduction is that the addition of the coupling agent destroys the original hydrogen bonds of the CNF and replaces them with new hydrogen bonds, resulting in a decrease in the crystallinity of the CNF. Simultaneously, the dual modification can destroy the hydrogen bonds among CNFs and greatly decrease the crystallinity. However, overall, the CNF modification process did not significantly reduce the crystallinity, even if the crystallinity was lower than bacterial cellulose, and the CNF still maintained its crystallinity. Therefore, the reinforcement effect of the CNF on materials would be maintained [35,43].

#### 3.1.4. Thermal Stability

Figure 6a,b shows the thermogravimetric (TG) and derivative thermogravimetric (DTG) curves of the CNF and modified CNF. The weight loss observed in the TG curves for all samples can be divided into three stages. In the first stage, the sample loses mass as the temperature reaches 200 °C, which is caused by the evaporation of water in the CNF (denoted by a small dehydration peak on the corresponding DTG curve). The second stage of weight loss occurs between 200 and 400 °C. For J-CNF, the mass loss at this stage is due to the decomposition of the silane coupling agent KH-550 (coated on the surface of CNF), cellulose, hemicellulose, lignin, and non-cellulose components; for Z-CNF, the weight loss at this stage is not only due to the decomposition of cellulose, hemicellulose, lignin, and non-cellulose components, but also due to the decomposition of l-malic acid branched onto CNF; and, for J-Z-CNF and Z-J-CNF, the mass loss at this stage is due to the decomposition of the silane coupling agent KH-550, l-malic acid, cellulose, hemicellulose, lignin, and non-cellulose components. The third stage occurs after 400 °C, and the mass loss at this stage might be related to the decomposition of lignin and the formation of ash.

As listed in Table 3, the initial decomposition temperature of the CNF is 251 °C, while that of J-CNF is 231 °C. *T*_1_ of J-CNF is lower than *T*_1_ of the CNF, because the modification of the CNF by the silane coupling agent KH-550 reduces the crystallinity of the CNF, and the thermal stability of CNF decreases. Compared with that of the CNF, the initial decomposition temperatures of Z-CNF and J-Z-CNF are lower at 201 °C and 208 °C, respectively. This phenomenon may be due to the following reasons: (1) the esterification reaction reduced the number of hydroxyl groups on the surface of the CNF, leading to a decrease in the intermolecular hydrogen bonding of the CNF; (2) there are more unstable carboxyl groups in Z-CNF and J-Z-CNF, which significantly reduce the thermal degradation temperature; and (3) esterification reduces the crystallinity of the surface structure of the modified CNF [39]. The initial decomposition temperature of Z-J-CNF is 273 °C, while the maximum decomposition temperature is 321 °C. The thermal stability of Z-J-CNF was greatly improved compared with that of the CNF, because although esterification can reduce the thermal stability of the CNF, the silanol produced by the hydrolysis of the coupling agent can combine with the hydroxyl groups on malic acid and the CNF to form new strong hydrogen bonds.

#### 3.1.5. Contact Angle Analysis

As shown in Figure 7, the contact angle of the CNF is 47 ± 1°, and the contact angles of J-CNF, Z-CNF, J-Z-CNF, and Z-J-CNF are 67° ± 2°, 74° ± 1°, 64° ± 3°, and 72° ± 2°, respectively. According to generally accepted definitions, when the contact angle between a surface and water is less than 90°, the surface is defined as hydrophilic; otherwise, the surface is hydrophobic. The contact angles of the modified CNFs are still less than 90°, but they are already very large compared to that of the unmodified CNF. The increase in the contact angle of J-CNF is because the silanol formed by the hydrolysis of the coupling agent combines with the hydroxyl groups on the surface of the CNF to generate new strong hydrogen bonds, thereby reducing the number of hydroxyl groups on the surface of the CNF. The proportion of C2 in the XPS spectrum decreases and the hydrophilicity of the surface of the CNF decreases. The reason for the increase in the contact angle of Z-CNF is the esterification reaction between L-malic acid and the hydroxyl groups on the surface of the CNF. The hydrophilic hydroxyl group on the surface of the CNF is replaced by the hydrophobic ester group, which improves hydrophobicity. Compared with cross-linking using the coupling agent, esterification with L-malic acid can increase the hydrophobicity of the CNF. The contact angle of Z-J-CNF is larger than that of J-Z-CNF, because of the greater influence of the first modification on the CNF, and esterification can better improve the hydrophobicity of the CNF.

### 3.2. Characterization of Cassava Residue CNF/Cassava Starch Composite Films

Among the four modified CNF types, Z-CNF has the smallest diameter, largest crystallinity, and largest contact angle. Meanwhile, Z-J-CNF has the best thermal stability and largest contact angle. Therefore, Z-CNF and Z-J-CNF were selected to improve the properties of cassava starch films.

#### 3.2.1. Mechanical Performance of Composite Films

As shown in Figure 8, with the increase in CNF content, the tensile strength of the composite shows an increasing trend first and then a decreasing trend, and reaches the peak value when the addition amount is 3%–4%. Compared with the original cassava starch film with a tensile strength of 2.4 ± 0.1 MPa, the tensile strength values of 3% CNF/TPS, 3% Z-CNF/TPS, and 3% Z-J-CNF/TPS nanocomposite films are 14.2 ± 1.7, 26.9 ± 1.0, and 24.6 ± 1.3 MPa, respectively, indicating increases of 500%, 1034%, and 936%, respectively. The addition of the CNF can significantly improve the mechanical strength of the cassava starch film. This research has the same trend as reported elsewhere [44]. Compared with the CNF, the enhancement effect of Z-J-CNF and Z-CNF is more obvious. This is mainly due to an improvement in the dispersibility of the modified CNF, which makes the CNF compatible with the cassava starch. In addition, the CNF with a high aspect ratio and fiber network structure can act as a fiber skeleton structure of the starch film, which greatly improves the tensile strength of the composite film [45]. At the same time, there is a good intermolecular interaction between the hydroxyl groups in the starch molecular chain and the carboxyl groups of Z-J-CNF and Z-CNF, which compacts the molecular structure of the composite film and further improves the tensile strength. With the increasing content in CNF, the tensile strength of the nanocomposites decreases to some extent. Because of the increase in the CNF content, its dispersibility in the starch film decreases, and it is easier for it to agglomerate.

However, the addition of the CNF and modified CNF lead to a downward trend in the elongation at break. This is mainly attributed to the binding of the CNF network structure and the strong interaction between the CNF and starch matrix, which restricts the movement of the starch molecular link. Therefore, the tensile strength of the composite film decreases significantly [46]. With the increase in the CNF content, the agglomeration phenomenon between the CNFs becomes obvious, the effect on the starch matrix gradually weakens, and the decrease in elongation at break becomes slow.

#### 3.2.2. Barrier Analysis

Figure 9 shows that the water vapor permeability (WVP) of pure cassava starch film is 3.3 ± 0.1 g·cm/(cm^2^·s·Pa), and the WVP of nanocomposite films with the CNF and modified CNF is lower than that of pure cassava starch film. The WVP of 3% CNF/TPS, 3% Z-CNF/TPS, and 4% Z-J-CNF/TPS nanocomposite films are 2.5 ± 0.1, 2.0 ± 0.1, and 2.1 ± 0.1 g·cm/(cm^2^·s·Pa), which are 25.68%, 35.05%, and 35.95% lower than that of the cassava starch film, respectively [47]. This is because cassava starch and the CNF, Z-CNF, or Z-J-CNF form strong hydrogen bonds, which stabilize the hydrophilic starch matrix. Z-CNF and Z-J-CNF have better compatibility with cassava starch than with the CNF, because Z-CNF and Z-J-CNF form stronger hydrogen bonds with the carboxyl and hydroxyl groups of starch, and the organic long chain of L-malic acid has a winding and stabilizing effect on starch [42]. When the amount of CNF and Z-CNF are greater than 3% and the amount of Z-J-CNF is greater than 4%, the WVP of the nanocomposite film decreases with the increase in the amount of the CNF and modified CNF. The agglomeration phenomenon of the CNF gradually becomes obvious with the increase in content. As a result, the adhesion between the reinforcement agent and the matrix is weakened. The microphase separation at the interface of the larger aggregates and starch also provides a path for the diffusion of water vapor [48].

As shown in Figure 10, the OP coefficient of pure cassava starch film is 0.91 ± 0.06 g∙mm∙m^−2^∙d^−1^. The addition of the CNF and modified CNF can obviously reduce the OP and improve the barrier of the nanocomposite membrane to oil. The minimum OP values of 3% CNF/TPS, 3% Z-CNF/TPS, and 3% Z-J-CNF/TPS nanocomposite films are 0.54 ± 0.03 g∙mm∙m^−2^∙d^−1^, 0.49 ± 0.03 g∙mm∙m^−2^∙d^−1^, and 0.46 ± 0.04 g∙mm∙m^−2^∙d^−1^; the OP of the films decreased by 40.66%, 46.15%, and 49.45%, respectively, compared with that of the cassava starch film. Z-CNF/TPS and Z-J-CNF/TPS nanocomposite films have better barrier properties to oils, mainly due to a better dispersibility of modified nanofibrils with better compatibility. With fewer microphase separation points, the structure of the nanocomposite film can be made tighter through the binding of the network structure, which reduces the ability of the lipid molecules to migrate. At the same time, the introduction of more hydrophilic carboxyl groups also hinders the diffusion of esters in the film to a certain extent, and the lipophilicity of Z-CNF/TPS and Z-J-CNF/TPS are relatively weaker. When less than 3% CNF and modified CNF are added, the OP coefficient decreases. However, as more CNF is added, the barrier properties of the film to oil are improved because the dense and continuous network structure of cellulose and starch molecules formed by hydrogen bond interactions can reduce the migration ability of oil molecules. When the amount of the added CNF and modified CNF is greater than 3%, the OP coefficient increases with added CNF, because, when the CNF and modified CNF are present at higher levels, the hydrogen bonding between the enhancers makes them unevenly dispersed in the starch system. This leads to agglomeration and the formation of many agglomerates, which makes the compatibility of the CNF and starch matrix worse, loosening the interface structure, and increasing the oil permeability coefficient of the film accordingly.

#### 3.2.3. Hydrophobicity of Composite Films

Figure 11 shows that CNF, Z-CNF, and Z-J-CNF make the cassava-starch-based nanocomposite films absorb less water than pure cassava starch films, and the water absorption of nanocomposite films with the modified CNF is lower than that of the nanocomposite with unmodified CNF. This is because some of the hydroxyl groups on the surface of Z-CNF and Z-J-CNF are esterified to form ester groups, which have a certain hydrophobic effect. At the same time, the hydroxyl and carboxyl groups in the Z-CNF and Z-J-CNF chains form strong hydrogen bonds with the hydroxyl groups in the cassava starch chain, which reduces the number of free hydroxyl groups of cassava starch [42]. The hygroscopicity of cassava-starch-based nanocomposite films also decrease as the amount of the enhancer (CNF, Z-CNF, and Z-J-CNF) is increased. This phenomenon can be explained by the following factors: (1) CNF, Z-CNF, and Z-J-CNF enhancers may not be as hygroscopic as starch due to their higher crystallinity; (2) the enhancers have good interfacial adhesion to starch, limiting the penetration and diffusion of moisture along the filler-enhancer interface; and (3) the presence of the enhancer increases the hardness and T_g_ of TPS and TG, and a network of CNFs is formed in the starch matrix, which reduces the fluidity of the starch chain, limiting the expansion of starch and reducing the water absorption [49,50].

As shown in Figure 12, the addition of the CNF can greatly increase the water contact angle of the composite film. The contact angle of the cassava starch film is 48° ± 3°. When the amount of the enhancer is 3%, the contact angles of the CNF/TPS, Z-CNF/TPS, and Z-J-CNF/TPS nanocomposites are 91° ± 2°, 107° ± 5°, and 111° ± 5°, respectively. Compared with that of the pure cassava starch film, the hydrophilicity of the enhanced film surface was reduced. This is due to the interactions between the components, and the generation of micro- and nanostructures on the surface of the film. Cassava starch segments expose their hydrophilic groups to reinforcing materials under new hydrogen bonding, exposing the hydrophobic groups to the surface and reducing the number of sites which can interact with water. The dispersed network structure of CNF also introduces a certain surface roughness to the composite film, which hinders the infiltration of water droplets on the surface, thereby increasing the hydrophobicity of the membrane [51]. When increased amounts of the CNF and modified CNF are added, the CNF and modified CNF agglomerate in the matrix and the contact angle decreases as a result. Compared with the contact angle of the CNF/TPS nanocomposite film, those of Z-CNF/TPS and Z-J-CNF/TPS are larger, because the dispersibility of the modified CNF in water is better, and the hydroxyl groups in the cassava starch chain are combined with the carboxyl groups and hydroxyl groups in Z-J-CNF and Z-CNF by chemical bonds. This can reduce the number of hydroxyl groups on the surface of Z-J-CNF and Z-CNF.

#### 3.2.4. Atomic Force Microscopy

To observe the changes in the surface roughness of the nanocomposite films, the prepared nanocomposite films were characterized by AFM (see Figure 13). The main reason for characterizing the nanocomposite films with 3% enhancer is that these nanocomposite films have better mechanical strength, barrier properties, and hydrophobicity. As shown in Figure 13, the surface of the pure starch film is relatively flat and smooth. With the addition of the CNF and modified CNF to the matrix, the surface of the film becomes uneven, the roughness of the nanocomposite increases, and irregular mastoid structures appear on the surface of the composite film, which induce the formation of micro- and nanostructures on the surface of the composite film. This is also an important reason for the significant increase in the contact angle of the composite film [52]. Figure 13 also shows that CNFs are evenly dispersed on the surface of the composite membrane without aggregation, indicating that the matrix and the CNFs in the composite film interact well. This result also verifies that the addition of the CNF can improve the mechanical properties of the film [53].

## 4. Conclusions

(1) In this study, four types of modified CNFs (J-CNF, Z-CNF, J-Z-CNF, and Z-J-CNF) were prepared using cassava residue CNF as the raw material and silane coupling agent KH-550 and l-malic acid as the modifiers. The effects of different modification methods on the physical and chemical properties of the CNFs were investigated. Compared with the unmodified CNF, the modified CNFs have significantly improved dispersibility, fibrils that are separated from each other, and they form three-dimensional network structures. No coarse fiber aggregates appeared. The diameter of Z-CNF decreased, while that of J-CNF increased when compared with that of the unmodified CNF. Modification with silane coupling agent KH-550 and l-malic acid decreased the crystallinity of CNFs, without affecting their crystal structure, which remained cellulose type 1 after modification. Among J-CNF, Z-CNF, J-Z-CNF, and Z-J-CNF, the contact angle of Z-CNF is the largest at 74° ± 1°. Z-J-CNF shows optimum thermal stability, with an initial decomposition temperature of 273°.

(2) When plasticized with glycerin and cast, a new non-toxic and environmentally friendly bio-nanocomposite film material was obtained from cassava starch and Z-CNF or Z-J-CNF as the reinforcement agent. The tensile strength of the cassava starch film was 2.4 ± 0.1 MPa, while those of 3% CNF/TPS, 3% Z-CNF/TPS, and 3% Z-J-CNF/TPS nanocomposite films were 14.2 ± 1.7, 26.9 ± 1.0, and 24.6 ± 1.3 MPa, higher by 500%, 1034%, and 936%, respectively, compared to that of the base film. The WVP of 3% CNF/TPS, 3% Z-CNF/TPS, and 3% Z-J-CNF/TPS composite films were reduced by 25.68%, 35.05%, and 35.95%, and the OP decreased by 40.66%, 46.15%, and 49.45%, respectively. The contact angle of the cassava starch film was 48° ± 3°, while those of 3% CNF/TPS, 3% Z-CNF/TPS, and 3% Z-J-CNF/TPS composite films were 91° ± 2°, 107° ± 5°, and 111° ± 5°, respectively.

(3) Z-CNF/TPS and Z-J-CNF/TPS composite films have higher tensile strength, better barrier properties, and lower hydrophilicity than the CNF/TPS composite film. The products obtained in this study are completely biodegradable, and the starch-based composite film filled with CNFs with high tensile strength and good barrier properties can considerably expand the applications of cassava residues.

## Figures and Tables

**Figure 1 nanomaterials-10-00755-f001:**
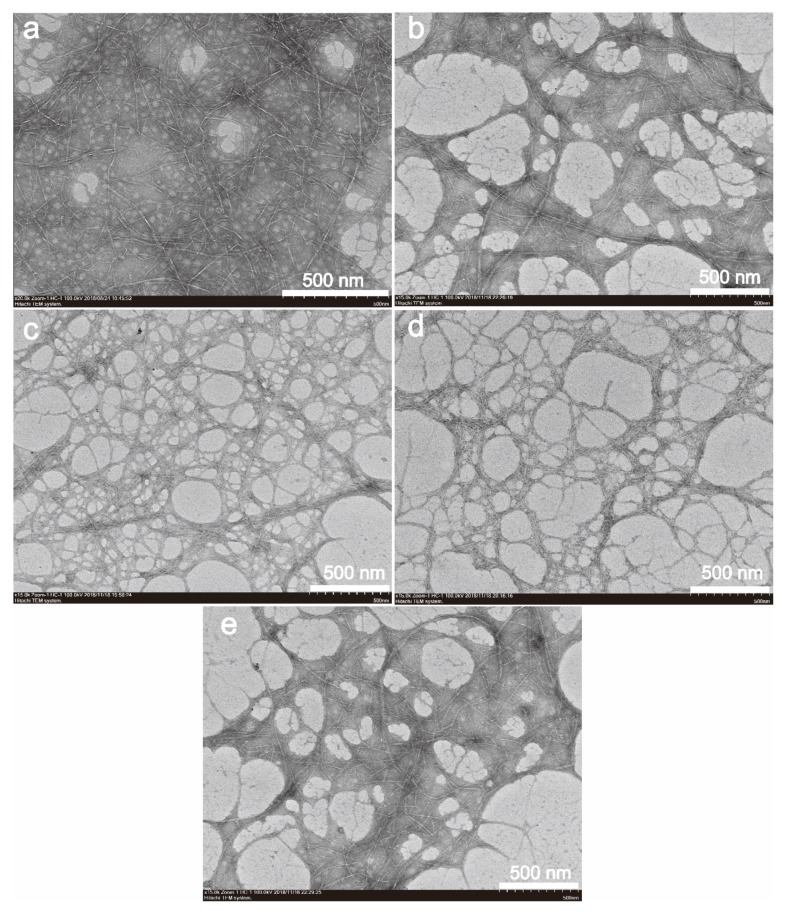
Transmission electron micrographs (TEM) of unmodified and modified cellulose nanofibril (CNF) samples: (**a**) CNF; (**b**) cross-linking (J)-CNF; (**c**) esterification (Z)-CNF; (**d**) J-Z-CNF; and (**e**) Z-J-CNF.

**Figure 2 nanomaterials-10-00755-f002:**
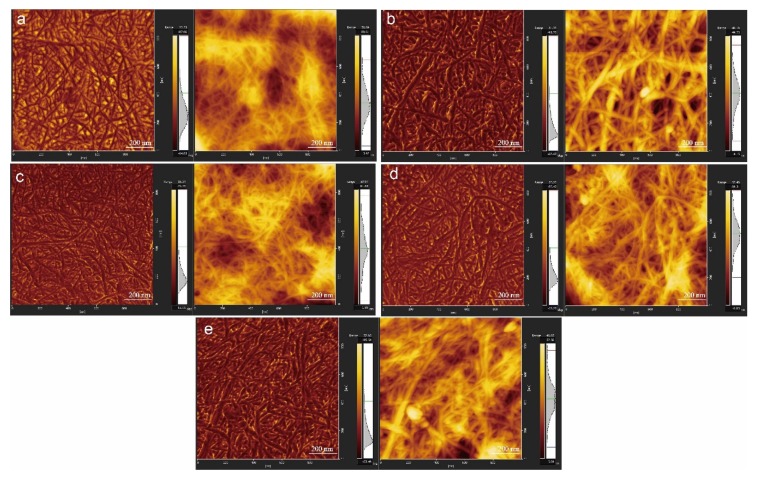
Atomic force microscopy (AFM) images of unmodified and modified CNF samples: (**a**) CNF; (**b**) J-CNF; (**c**) Z-CNF; (**d**) J-Z-CNF; and (**e**) Z-J-CNF.

**Figure 3 nanomaterials-10-00755-f003:**
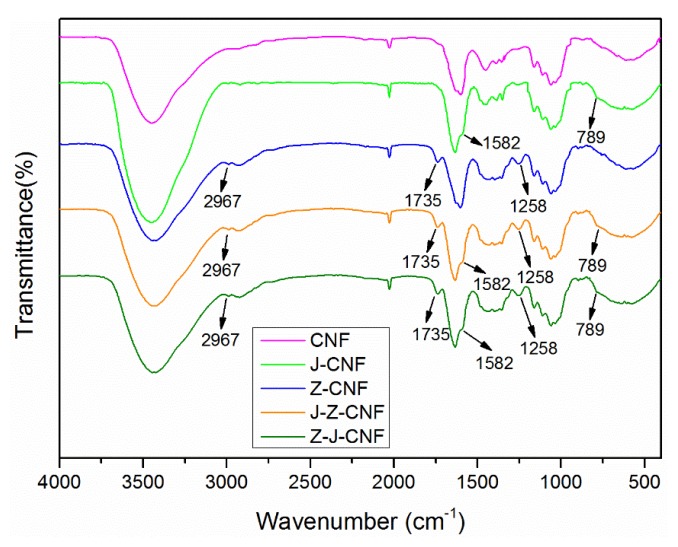
Fourier-transform infrared spectrometry (FT-IR) spectra of CNF and modified CNF samples.

**Figure 4 nanomaterials-10-00755-f004:**
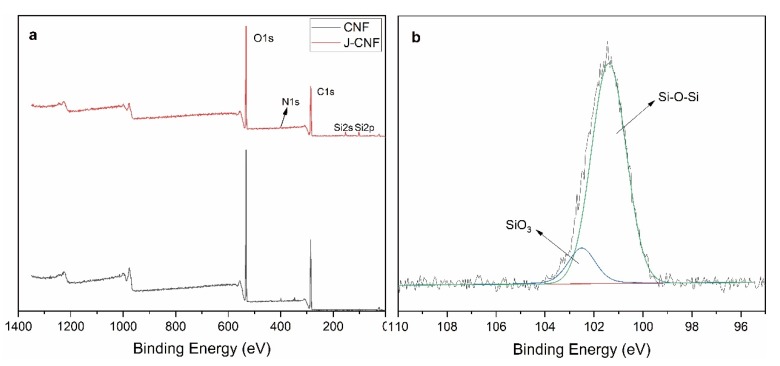
X-ray photoelectron spectroscopy (XPS) spectra of CNF and J-CNF: (**a**) wide scan spectra; and (**b**) Si2p spectrum of J-CNF.

**Figure 5 nanomaterials-10-00755-f005:**
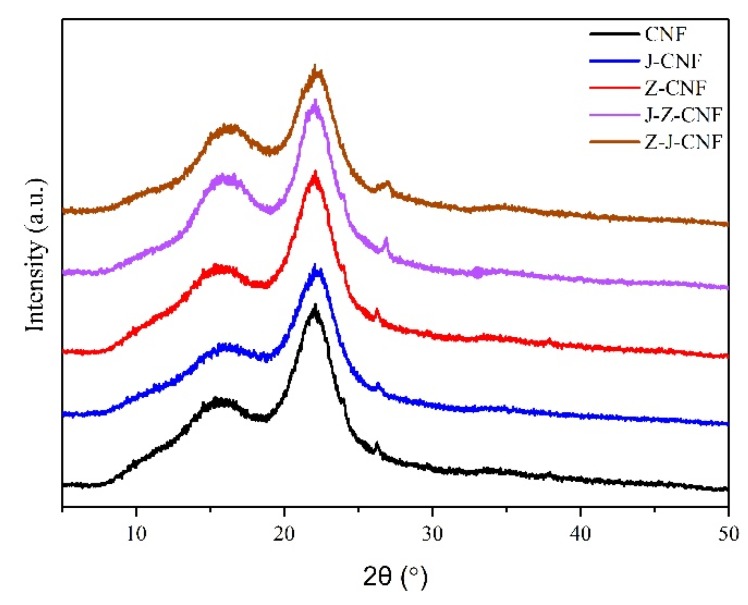
X-ray diffraction (XRD) patterns of CNF and modified CNF.

**Figure 6 nanomaterials-10-00755-f006:**
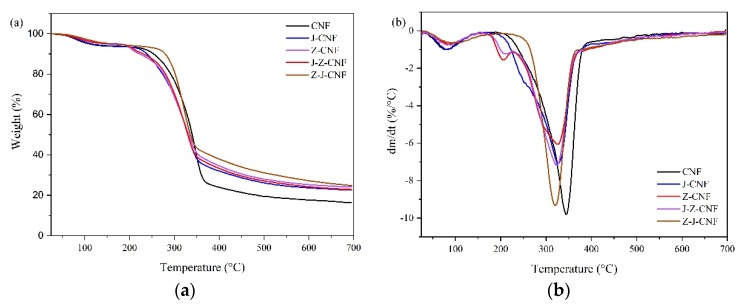
(**a**) thermogravimetric (TG); and (**b**) derivative thermogravimetric (DTG) curves of the CNF and modified CNF.

**Figure 7 nanomaterials-10-00755-f007:**
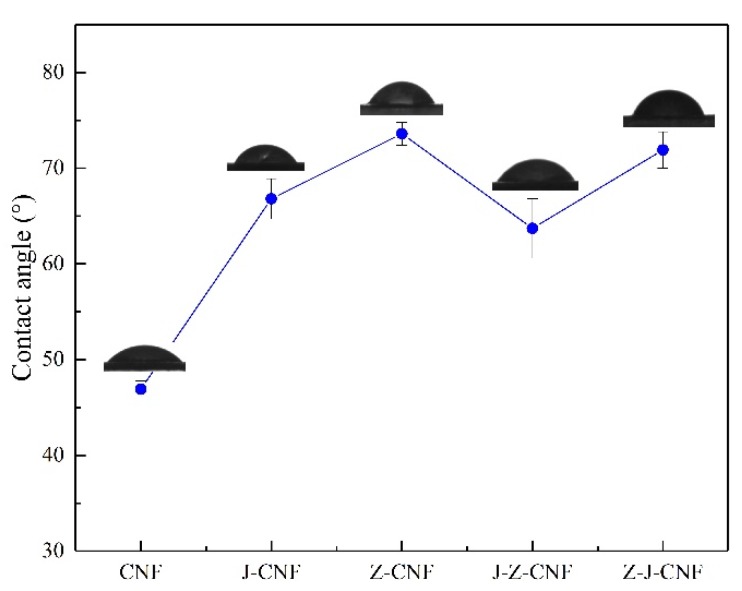
Contact angles of the CNF and modified CNF.

**Figure 8 nanomaterials-10-00755-f008:**
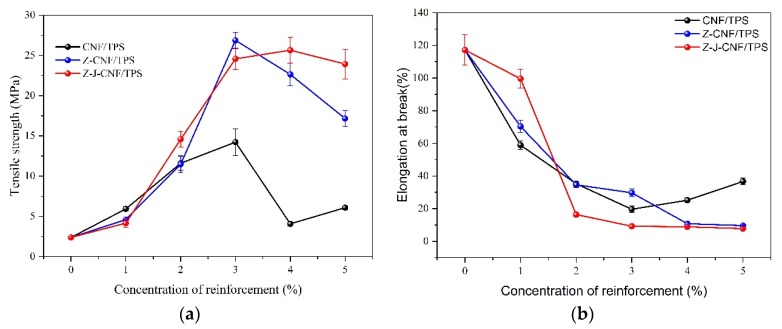
Effects of CNF, Z-CNF, and Z-J-CNF content of the composite films on: (**a**) the tensile strength; and (**b**) the elongation at break.

**Figure 9 nanomaterials-10-00755-f009:**
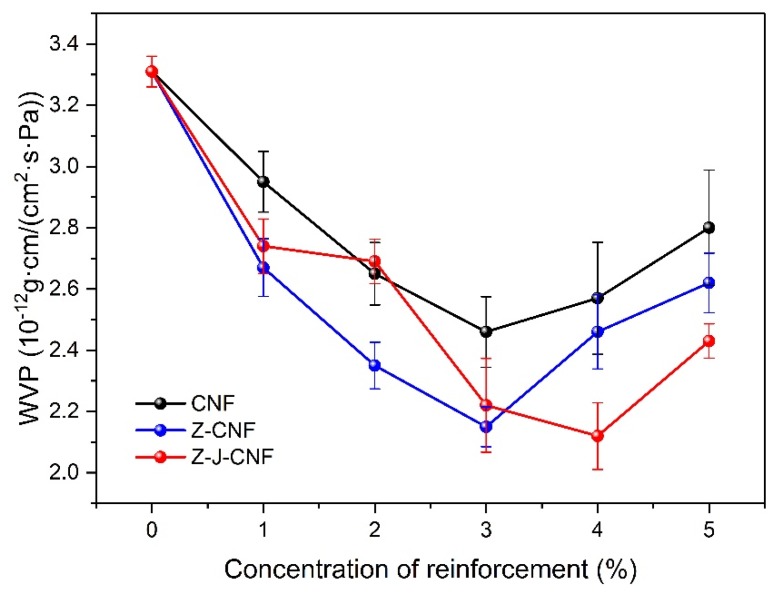
Effects of CNF, Z-CNF, and Z-J-CNF content on the water vapor permeability (WVP) of the films.

**Figure 10 nanomaterials-10-00755-f010:**
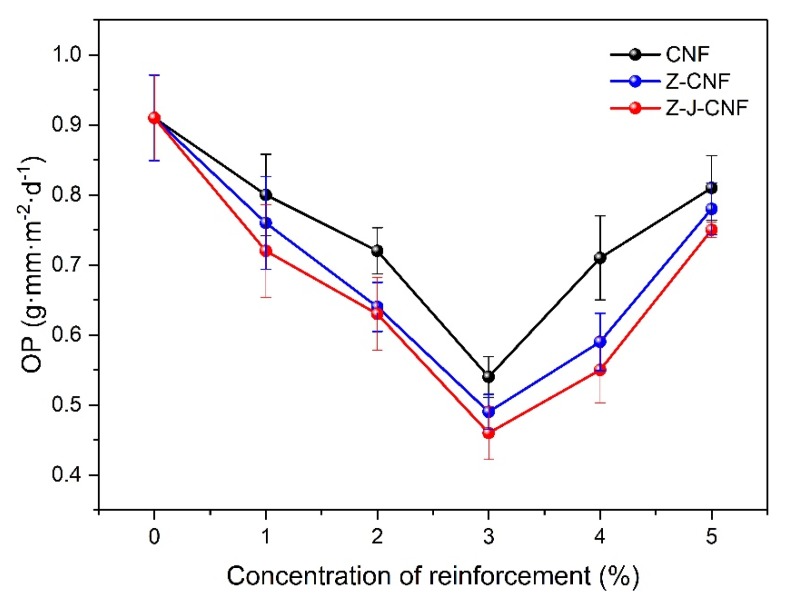
Effects of CNF, Z-CNF, and Z-J-CNF content on oil permeability (OP) of the films.

**Figure 11 nanomaterials-10-00755-f011:**
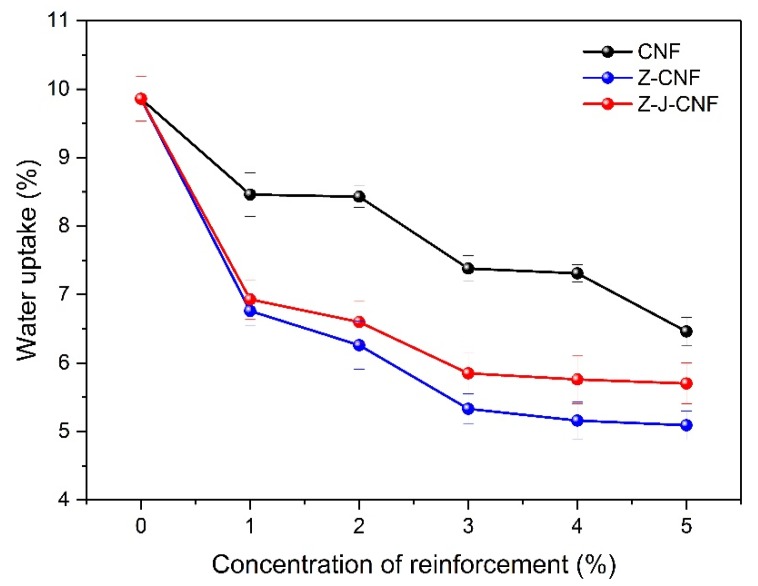
Effects of CNF, Z-CNF, and Z-J-CNF content on water absorption of the films.

**Figure 12 nanomaterials-10-00755-f012:**
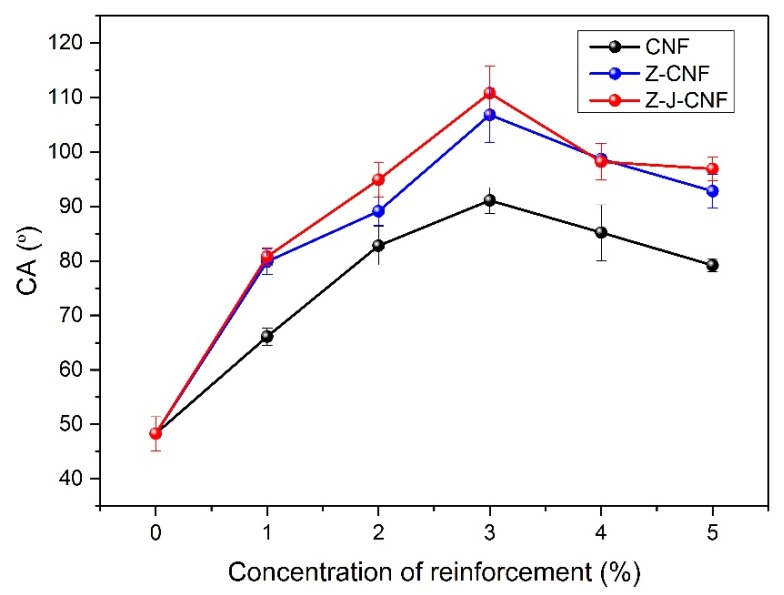
Effects of CNF, Z-CNF, and Z-J-CNF content on the contact angle of the films.

**Figure 13 nanomaterials-10-00755-f013:**
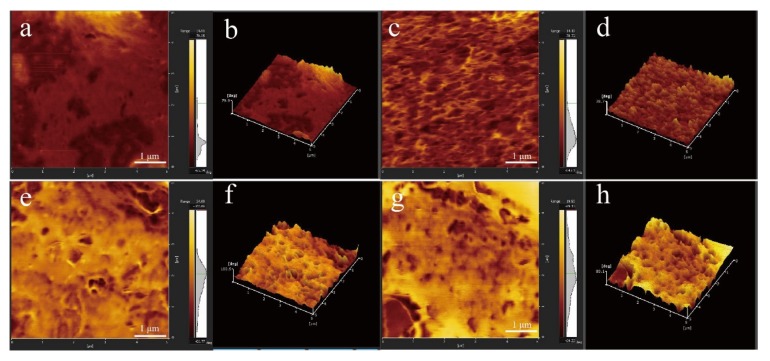
AFM images of cassava starch film and nanocomposite film: (**a**) cassava starch film; (**b**) 3D image of (**a**); (**c**) 3% CNF/ thermoplastic cassava starch (TPS) nanocomposite film; (**d**) 3D image of (**c**); (**e**) 3% Z-CNF/TPS nanocomposite film; (**f**) 3D image of (**e**); (**g**) 3% Z-J-CNF/TPS nanocomposite film; and (**h**) 3D image of (**g**).

**Table 1 nanomaterials-10-00755-t001:** Relative distribution of carbon atoms of cellulose nanofibril (CNF) and cross-linking (J)-CNF.

Sample	C1 (C–C)	C2 (C–OH)	C3(O–C–O)
CNF	25% ± 2%	47% ± 4%	28% ± 3%
J-CNF	33% ± 2%	38% ± 3%	28% ± 2%

**Table 2 nanomaterials-10-00755-t002:** Crystallinity index (*Cr*_I_) of CNF and modified CNF.

Sample	Crystallinity/%
CNF	57.5 ± 2.0
J-CNF	53.2 ± 4.2
Z-CNF	55.4 ± 4.3
J-Z-CNF	53.1 ± 2.8
Z-J-CNF	50.3 ± 3.1

**Table 3 nanomaterials-10-00755-t003:** Degradation temperatures and weight loss of the CNF and modified CNF based on derivative thermogravimetric (DTG) curves.

Sample	*T*_1_/°C	Weight Loss/%	*T*_max_/°C	Weight Loss/%	At 700 °C Residual Amount/%
CNF	251	9.6 ± 0.9	343	54.0 ± 3.5	16.3 ± 1.2
J-CNF	231	8.7 ± 0.6	326	47.8 ± 4.3	22.6 ± 2.0
Z-CNF	201	8.0 ± 0.8	326	46.4 ± 5.6	24.2 ± 2.9
J-Z-CNF	208	7.6 ± 0.5	324	45.4 ± 4.4	22.9 ± 1.8
Z-J-CNF	273	8.7 ± 0.4	321	36.9 ± 2.7	24.8 ± 2.2

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
