# Peer review of "Preparation and Properties of Cassava Residue Cellulose Nanofibril/Cassava Starch Composite Films"

_nanomaterials, 2020, doi:10.3390/nano10040755_

Round 1

Reviewer 1 Report

It is an interesting contribute in the field. Results are interesting and conclusion supported by experiments.

I suggest the following minor revision before publication.

  • Fig 2 and 13 labels are very small. Please use a larger font size.
  • “weight loss” should be “weight %” or “mass % “. The initial value is 100%.
  • What is “Quality loss” in table 3?

Reviewer 2 Report

These authors have prepared a complex cellulosic/starch composite interlinked with a silicone coupling agent and malic acid which had good physical and chemical properties with high thermal stability as an environmentally friendly packaging film material. The outcomes look interesting and are well underpinned by a wide range of physical and chemical methodologies. The grammar nedds very minor attention although not serious. For myself an interesting article.
